# Language-Related Disparities in Pain Management in the Post-Anesthesia Care Unit for Children Undergoing Laparoscopic Appendectomy

**DOI:** 10.3390/children7100163

**Published:** 2020-10-04

**Authors:** Anjali A. Dixit, Holly Elser, Catherine L. Chen, Marla Ferschl, Solmaz P. Manuel

**Affiliations:** 1Department of Anesthesiology and Pain Medicine, Seattle Children’s Hospital, University of Washington, Seattle, WA 98105, USA; anjalid@uw.edu; 2School of Medicine, Stanford University, Stanford, CA 94309, USA; hollys1@stanford.edu; 3Department of Anesthesia and Perioperative Care, University of California, San Francisco, CA 94143, USA; catherine.chen@ucsf.edu (C.L.C.); marla.ferschl@ucsf.edu (M.F.)

**Keywords:** pediatric acute pain, perioperative care, general surgery, ambulatory surgery, anesthesiology, healthcare disparities, minority health, language

## Abstract

Race and ethnicity are associated with disparities in pain management in children. While low English language proficiency is correlated with minority race/ethnicity in the United States, it is less frequently explored in the study of health disparities. We therefore investigated whether English language proficiency influenced pain management in the post-anesthesia care unit (PACU) in a cohort of children who underwent laparoscopic appendectomy at our pediatric hospital in San Francisco. Our primary exposure was English language proficiency, and our primary outcome was administration of any opioid medication in the PACU. Secondary outcomes included the amount of opioid administered in the PACU and whether any pain score was recorded during the patient’s recovery period. Statistical analysis included adjusting for demographic covariates including race in estimating the effect of language proficiency on these outcomes. In our cohort of 257 pediatric patients, 57 (22.2%) had low English proficiency (LEP). While LEP and English proficient (EP) patients received the same amount of opioid medication intraoperatively, in multivariable analysis, LEP patients had more than double the odds of receiving any opioid in the PACU (OR 2.45, 95% CI 1.22–4.92). LEP patients received more oral morphine equivalents (OME) than EP patients (1.64 OME/kg, CI 0.67–3.84), and they also had almost double the odds of having no pain score recorded during their PACU recovery period (OR 1.93, CI 0.79–4.73), although the precision of these estimates was limited by small sample size. Subgroup analysis showed that children over the age of 5 years, who were presumably more verbal and would therefore undergo verbal pain assessments, had over triple the odds of having no recorded pain score (OR 3.23, CI 1.48–7.06). In summary, English language proficiency may affect the management of children’s pain in the perioperative setting. The etiology of this language-related disparity is likely multifactorial and should be investigated further.

## 1. Introduction

Disparities in pediatric pain management associated with race/ethnicity are well-documented. In emergency departments nationwide, Black children receive opioids at significantly lower rates than White children when presenting with acute appendicitis [1]. In the outpatient setting, White children are more likely than non-White children to receive opioid prescriptions for pain management [2]. However, data in the perioperative environment remain limited with mixed findings: Jimenez and colleagues found that children of Latin American descent receive opioids in the post-anesthesia care unit (PACU) at lower rates following tonsillectomy and adenoidectomy [3]. In contrast, Naifu and colleagues found that non-White children were more likely to be administered intravenous opioids than White children [4].

In the United States, race and language are intricately connected, with scholars describing the practice of *linguicism* (i.e., discrimination based on primary spoken (non-English) language) and expressing that “language is a social practice that shapes subjectivity and establishes power relations among members of different racial and class groups” [5]. English language proficiency, which may be correlated with race/ethnicity [6,7], is however less frequently considered in the study of health disparities among children. Language barriers have been described as increasing risk of adverse events for adult patients throughout the hospital encounter [8]. In addition, non-White adults who are not proficient in English are less likely to receive empathy, establish rapport, benefit from adequate communication, and be involved in shared decision-making with their physicians [7]. In the pediatric population, parents of children whose primary language is not English are more likely to be dissatisfied with the amount of time and quality of care provided by their children’s clinicians [9]. In addition, children whose families have low English proficiency receive fewer postoperative daily pain assessments and require higher pain scores before receiving opioid medications [10].

The present study is focused on further examining the role of language disparities in perioperative care. Specifically, we conducted a retrospective cohort study in order to understand whether English language proficiency influenced PACU pain management in a cohort of children who underwent laparoscopic appendectomy at an urban pediatric hospital in San Francisco, where a large proportion of residents do not speak English at home [11]. We hypothesized that children with low English proficiency would be less likely to receive opioid analgesia in the PACU compared to children who were proficient in English given their limited ability to verbally establish rapport and communicate their pain to their caregivers.

## 2. Materials and Methods

### 2.1. Data Sources and Study Cohort

We used electronic medical record data from our academic pediatric hospital located in Northern California to conduct a retrospective cohort study among patients who underwent laparoscopic appendectomy between February 2015 and July 2019. Eligible patients were age 18 years or younger and were recovered in the PACU after surgery. All laparoscopic appendectomies were performed by pediatric general surgeons who used three port sites and administered local anesthetic subcutaneously at these sites during surgical closure. All anesthetics were administered by pediatric anesthesiologists who provided a balanced anesthetic with a volatile agent, a medium- to long-acting opioid (fentanyl, hydromorphone, and/or morphine), and boluses or infusion of an intravenous sedative/hypnotic (propofol). Other analgesics and sedative/hypnotics including short-acting opioids (remifentanil, alfentanil, and sufentanil), ketamine, and dexmedetomidine were not routinely used. For each patient, follow-up extended from the four hours preceding surgery until they were discharged from the PACU.

### 2.2. English Proficiency

The primary exposure of interest in our analysis was English language proficiency, which we determined based on whether an interpreter was needed when the patient and their parent or guardian presented for their appendectomy or for any prior clinical encounter. Patients were categorized as having low-English-proficiency (LEP) if they indicated that an interpreter was needed and designated as English-proficient (EP) otherwise.

### 2.3. Opioid Medication

Our primary outcome of interest was whether a patient received any opioid medication in the PACU. Opioid medications included fentanyl, morphine, hydromorphone, oxycodone, hydrocodone, methadone, and tramadol. We created an indicator variable that equaled one if the patient received any of these opioid medications in the PACU and zero otherwise. Secondary outcomes included the amount of opioid administered in the PACU, calculated in oral morphine equivalents (OME) per kilogram [12,13]. We also created an indicator variable that equaled one if any pain score (on a scale of 0–10) was recorded in the PACU and equaled zero if no pain score was recorded by the patient’s nurse. Nurses in the PACU at our institution quantify pain using the Face, Legs, Activity, Cry, Consolability (FLACC) [14] scale in pre-verbal children and using a numerical (0–10) or descriptive (mild, moderate, severe) scale that is converted to a number in verbal children.

### 2.4. Covariates

Patient demographics included in the present study were continuous age (0–18); categorical patient- or family-reported gender (male, female, other); categorical patient- or family-reported ethnicity (Hispanic or Non-Hispanic); and categorical patient- or family-reported race. For race assignation, patients and their parent/guardian could decline to answer or self-identify one or more categories from the U.S. Census list of race categories (White, Black or African American, American Indian or Alaska Native, Asian, and Native Hawaiian or Other Pacific Islander) [15], “Other,” and “Unknown.” Given the small numbers, we categorized patients who self-identified as one of the following categories into the “Other” category: individuals who identified as multiracial by selecting more than one of these categories, those who identified as Native Hawaiian or Other Pacific Islander, American Indian or Alaska Native, and those who declined to answer or selected “Unknown”.

We also compared LEP and EP patients in terms of American Society of Anesthesiologist ratings, preexisting medical conditions (asthma, obstructive sleep apnea, cough, pneumonia, obesity, cancer, sepsis, diabetes, and developmental delay), use of opioids before admission, time in the operating room, presence of appendiceal perforation and/or peritonitis, opioids administered intraoperatively (in OME per kilogram), nonopioid adjunct medications administered intraoperatively (acetaminophen and/or ketorolac), and occurrence in the PACU of laryngospasm, postoperative nausea/vomiting, low respiratory rate (<10 breaths per minute), and low oxygen saturation (<90%). We did not include these variables in our subsequent statistical analyses because we hypothesized that they were related to the outcomes in the causal pathway solely as mediators, rather than as confounders.

### 2.5. Statistical Analysis

We used generalized logistic regression to estimate the association between language proficiency and PACU opioid administration. We hypothesized that demographic factors may influence how PACU nurses assessed and treated patients’ pain. This model was therefore adjusted for demographic characteristics as enumerated above. Ethnicity was not included in this model or subsequent models as only three patients self-identified as Hispanic.

Next, we examined the association between language proficiency and the amount of opioid administered in the PACU in OME per kilogram. The observed distribution of OME per kilogram administered exhibited overdispersion due to a point mass at zero and right skew, which we accommodated by using negative binomial regression, adjusting for demographic characteristics as described above. We also conducted a supplementary analysis using this method to estimate the association between language proficiency and the amount of intraoperative opioid administered in OME per kilogram.

Finally, we used generalized logistic regression to estimate the association between language proficiency and whether a pain score was incorporated in the PACU after adjusting for demographic characteristics. We then conducted a subgroup analysis by stratifying our cohort by age (age less than 5 years, or greater than or equal to 5 years) to delineate differences in how nurses may interact with and elicit pain scores from young children versus older, school-aged, more verbal children.

Statistical analyses were performed using R version 3.6.3 (R Foundation for Statistical Computing, Vienna, Austria) [16]. This study was approved by the University of California, San Francisco Institutional Review Board, protocol 19-28181.

## 3. Results

During our study period, 257 pediatric patients underwent laparoscopic appendectomy and subsequently recovered in the PACU. Of these, 57 (22.2%) were identified as LEP. The most common language spoken by LEP patients was Spanish (80.7%). Characteristics of these patients are presented in Table 1. All patients received general anesthesia, and none received supplemental regional or neuraxial anesthesia. One patient received dexmedetomidine intraoperatively; no patients received remifentanil, alfentanil, sufentanil, or ketamine. On average, LEP patients were about 2 years younger than EP patients. LEP patients were more likely to select “Other” as their race (87.7% compared to 41.5%) and to select any non-White race (91.2% compared to 53.5%). The vast majority (98.8%) of patients selected Non-Hispanic as their ethnicity. Less than half of the patients had any medical comorbidity. LEP and EP patients did not have any notable differences in perioperative characteristics, including laryngospasm, postoperative nausea/vomiting, low respiratory rate or oxygen saturation in the PACU, presence of localized or generalized peritonitis, or appendiceal perforation. LEP children were more likely to receive two nonopioid adjuncts (acetaminophen and ketorolac) intraoperatively. Notably, there was no difference in intraoperative opioid administration (in OME/kg) between English proficiency groups, in either unadjusted analysis or subsequent negative binomial regression analysis adjusting for demographic characteristics (Appendix A).

For the remainder of the results, we report 95% confidence intervals but do not interpret results using null hypothesis significance testing. Rather, we attempt to determine if regression results imply an underlying relationship between exposure and outcome [17].

In multivariable analysis, LEP patients had more than double the odds of being administered any opioid in PACU compared to EP patients (OR 2.45, 95% CI 1.22–4.92) (Table 2). LEP patients also received more OME per kilogram than EP patients (1.64, CI 0.67–3.84) (Table 3). The estimated effect of being LEP was approximately 1.64 OME per kilogram: in other words, a 10-kg LEP child would receive approximately 16.4 more OME compared to an EP child after accounting for covariates. The precision of this estimate of OME was limited, as reflected by wide confidence bounds.

LEP patients also had approximately double the odds of having no pain score recorded during their PACU recovery periods (OR 1.93, CI 0.79–4.73) (Table 4). In subgroup analysis stratifying children by school-age (≥5 years) or not school-age (<5 years), older children who were LEP had over triple the odds of having no pain score recorded (OR 3.23, CI 1.48–7.06) (Table 5). Younger children who were LEP had lower odds, although this estimate was relatively imprecise given the small number of patients included in this subgroup (OR 0.5, CI 0.04–6.51). Of note, 82 patients in the total study sample had no pain score recorded, but 33 of them (40.2%) were administered opioid medication in the PACU.

## 4. Discussion

Our retrospective cohort study identifies English language proficiency as a potential contributor to disparities in children’s perioperative pain management, in addition to the racial disparities that have been reported previously in the PACU [4,18] and the language disparities reported in the post-surgical wards [10]. Based on prior studies, we hypothesized that LEP children would be less likely to receive opioid analgesia than their EP counterparts. However, in our cohort, LEP children had higher odds of receiving opioid analgesia in the PACU compared to EP children despite receiving the same amount of intraoperative opioid analgesia and higher numbers of intraoperative nonopioid analgesics (i.e., acetaminophen and ketorolac). LEP children, particularly those over 5 years old and therefore more likely to undergo a verbal, question-based pain assessment that might require an interpreter (rather than a non-verbal, face-scale or other clinical assessment [19,20]), were also more likely not to undergo any objective pain assessment in the PACU. Approximately 40% of patients without a documented pain score still received opioid analgesia.

Optimal treatment of pain requires balancing both objective and subjective assessments of pain and can prevent long-lasting complications in a post-surgical population. Overreliance on objective pain scales, without incorporation of clinicians’ and patients’ subjective assessments, can increase patient anxiety and diminish the therapeutic alliance among patients, family members, and their healthcare providers [21]. However, overreliance on subjective assessments can introduce conscious or unconscious bias into a clinician’s assessment, as has been documented in studies investigating racial [22] and gender-based [23] prejudice in the treatment of pain. When children’s acute procedural or surgical pain is undertreated, it can lead to long-term effects [24] including the development of fear and anxiety surrounding medical procedures into adulthood [25], changes in sensitivity and development of hyperalgesia [26] or persistent post-surgical pain [27], and alterations in nociceptive pathways [28]. On the other hand, overtreatment can lead to oversedation and respiratory depression, which can in turn result in significant morbidity and mortality [29]. Studies in North America and in Europe have demonstrated that severe respiratory and cardiac events in the perioperative period—including in the PACU—are prevalent in the pediatric surgical population [30,31]. The APRICOT trial, which pooled prospective data from 33 European countries, found that approximately 5% of anesthetics in pediatric patients resulted in severe cardiopulmonary events [31]. Secondary analyses of these data revealed that centers or anesthesiologists more specialized in pediatric care were less likely to report these adverse events [32], implying that those with more experience with children and their unique physiology may be better equipped to prevent events such as drug overdose resulting in oversedation, respiratory failure, and/or cardiac arrest.

In our cohort, a large proportion of children underwent no objective measurement of pain during their PACU recovery period, with LEP children being more likely to have no recorded pain score and more likely to receive opioid medication in the PACU. Our data did not elucidate what types of subjective measurements clinicians may have used to assess pain before administering opioids, or whether LEP children were overtreated or oversedated. While there were almost no documented occurrences of nausea/vomiting, low respiratory rate, or low oxygen saturation in the PACU in either the LEP or EP groups, these outcomes are often transient and may not be adequately captured in our medical record. In our open-model PACU, a low oxygen saturation event may be shorter than a minute and be rapidly corrected by stimulation or administration of supplemental oxygen. Such transitory events would not automatically be entered into the patient’s medical record. Future studies should investigate the types of non-objective assessments being done in the PACU and whether these do indeed result in oversedation.

The reasons behind language-related differences in care—including the likelihood not to use objective measures of pain—are likely multifactorial and related both to interpersonal and systemic factors. Interpersonal factors may include conscious or subconscious bias against those who do not speak English (i.e., seeing LEP patients as outsiders, with their pain as less relatable) [6]; patients’ or clinicians’ differing cultural expectations or beliefs related to the experience of pain [33], with language serving as a proxy for those cultural beliefs; lower levels of health literacy and knowledge on the part of LEP parents to advocate for their child [6]; the perceived additional time and effort it would take to objectively assess a child’s pain and pacify them when a language barrier exists. On the last point, a clinician may perceive it to be easier or less time-consuming to treat a non-English-speaking child with medication rather than nonpharmacologic methods. Systemic factors that may exacerbate these differences include lack of multilingual staff, lack of access by medical professionals to language classes in commonly spoken non-English languages [34], and interpreter services being difficult to access, particularly in the perioperative setting and at times when emergency surgery may occur. All of these factors could potentially be modified to attenuate the differences found between LEP and EP patients’ PACU pain management.

### Limitations

This retrospective study was based on language assessment data from inpatient and outpatient admissions personnel and may therefore be subject to selection or information bias. A bilingual child may have been denoted as LEP if their parent or guardian was LEP; thus, a nurse could potentially assess the patient’s pain in English in the PACU even if the parent or guardian may have needed interpretation for more complicated discussions. Further, our medical record does not include information on non-English languages spoken by the patient’s assigned nurse. These types of errors are common in language data collection [35], and in this study they would most likely have diluted the differences we calculated between LEP and EP patients. Currently, our hospital system only routinely collects interpreter data on admission, and a video interpreter services is available at all times, but clinicians are not required to document if and when they call on an interpreter during the hospital encounter except during procedural consent discussions. Language data collection could be improved [8] to generate more precise assessments in the future.

Children with generalized peritonitis and/or perforation experience more pain following laparoscopic appendectomy [36]. While our dataset did not include preoperative vital signs, laboratory values, or Alvarado score [37], which would have served as preoperative indicators of severity of appendicitis and may have correlated with postoperative pain, we were able to identify patients with peritonitis and/or perforation based on diagnosis codes and problem lists in our electronic medical record. Based on these data the occurrence of more severe cases of appendicitis did not differ between language proficiency groups; thus, we do not believe that our findings are skewed by severity of illness.

Our data also illustrate the complexity in portraying constructs such as ethnicity. Only three of 54 Spanish-speaking patients self-identified as Hispanic, findings which are inconsistent with our city demographics [11]. Self-identified race and ethnicity reflect changing social mores, connotations, and sociopolitical paradigms, and these labels do not fully capture how different groups of people may experience inequality [38]; our data on ethnicity reflects this dissonance.

Finally, our study was conducted at a single academic pediatric hospital in San Francisco, and findings therefore may be unlikely to reflect the experiences of patients who receive care in other hospital systems due to meaningful differences in patient and staff demographics. Subsequent larger studies may provide more precise and generalizable findings regarding potential language-related biases in pediatric populations.

## 5. Conclusions

In this retrospective cohort study of 257 pediatric patients, we found that English language proficiency may affect the management of children’s pain in the perioperative setting at our institution. Our study underscores the importance of communication in decisions surrounding pain management. Our findings motivate further investigation regarding the role that language plays in treatment decisions in pediatric patient populations, as well as efforts to identify strategies to optimize pain management for all of our patients regardless of their language.

## Figures and Tables

**Table 1 children-07-00163-t001:** Characteristics of the study population.

	All Patients	English Proficient (EP)	Low English Proficient (LEP)
N (%)	257	200 (77.8)	57 (22.2)
**Demographics**			
Age in years (Mean, SD)	10.14 (4.1)	10.63 (3.9)	8.44 (4.3)
Gender (N (%))			
*Male*	149 (58.0)	114 (57.0)	35 (61.4)
*Female*	109 (42.0)	86 (43.0)	22 (38.6)
Primary Language (N (%))			
*English*	194 (75.5)	189 (94.5)	5 (8.8)
*Spanish*	54 (21.0)	8 (4.0)	46 (80.7)
*Other*	9 (3.5)	3 (1.5)	6 (10.5)
Ethnicity (Non-Hispanic) (N (%))	254 (98.8)	197 (98.5)	57 (100.0)
Race (N (%))			
*White*	98 (38.1)	93 (46.5)	5 (8.8)
*Black*	8 (3.1)	8 (4.0)	0 (0.0)
*Asian*	18 (7.0)	16 (8.0)	2 (3.5)
*Other*	133 (51.8)	83 (41.5)	50 (87.7)
Any Non-White Race Selected (N (%))	159 (61.9)	107 (53.5)	52 (91.2)
**Clinical Characteristics**			
ASA Class (N (%))			
*1*	157 (62.3)	127 (64.8)	30 (53.6)
*2*	83 (32.9)	62 (31.6)	21 (37.5)
*3*	12 (4.8)	7 (3.6)	5 (8.9)
Medical Comorbidities (N (%))			
*Opioids on Admission Medication List*	9 (3.5)	5 (2.5)	4 (7.0)
*Asthma*	21 (8.2)	13 (6.5)	8 (14.0)
*Obstructive Sleep Apnea*	6 (2.3)	6 (3.0)	0 (0.0)
*Cough*	3 (1.2)	2 (1.0)	1 (1.8)
*Pneumonia*	2 (0.8)	2 (1.0)	0 (0.0)
*Obesity*	17 (6.6)	15 (7.5)	2 (3.5)
*Cancer*	2 (0.8)	2 (1.0)	0 (0.0)
*Sepsis*	3 (1.2)	2 (1.0)	1 (1.8)
*Diabetes*	1 (0.4)	1 (0.5)	0 (0.0)
*Developmental Delay*	1 (0.4)	0 (0.0)	1 (1.8)
**Perioperative Characteristics**			
OME, in 4 h pre-OR (Mean (SD))	4.07 (7.8)	4.20 (8.1)	3.63 (7.1)
OME, Intraoperative (Mean (SD))	24.15 (17.3)	24.68 (17.4)	22.26 (16.9)
Nonopioid Adjuncts, Intraoperative			
0	13 (5.1)	12 (6.1)	1 (1.8)
1	207 (81.2)	163 (82.3)	44 (77.2)
2	35 (13.7)	23 (11.6)	12 (21.1)
Minutes in OR (Mean (SD))	85.25 (20.2)	84.55 (20.1)	87.68 (20.7)
Minutes in PACU (Mean (SD))	102.58 (46.3)	99.63 (45.0)	112.89 (49.5)
Laryngospasm (N (%))	0 (0.0)	0 (0.0)	0 (0.0)
Postoperative Nausea/Vomiting (N (%))	1 (0.4)	1 (0.5)	0 (0.0)
Low Respiratory Rate (N (%))	2 (0.8)	2 (1.0)	0 (0.0)
Low Oxygen Saturation (N (%))	3 (1.2)	3 (1.5)	0 (0.0)
Peritonitis, Localized (N (%))	69 (28.6)	52 (26.0)	17 (29.8)
Peritonitis, Generalized (N (%))	37 (14.4)	26 (13.0)	11 (19.3)
Appendiceal Perforation (N (%))	17 (6.6)	14 (7.0)	3 (5.3)

Abbreviations: ASA, American Society of Anesthesiologists; OME, oral morphine equivalents; OR, operating room; PACU, post-anesthesia care unit

**Table 2 children-07-00163-t002:** Odds of opioid administration in the PACU, by English proficiency.

	Adjusted OR (95% CI) *
Low English proficient (ref = English proficient)	2.45 (1.22, 4.92)
Age in years	1.14 (1.06, 1.22)
Gender (ref = male)	0.6 (0.35, 1)

Abbreviations: OR, odds ratio; CI, confidence interval; * Coefficients were also adjusted for race (White, Black, Asian, and other) but these are not reported given the small cohort size of some racial subgroups.

**Table 3 children-07-00163-t003:** Opioid administration (in OME per kilogram) in the PACU, by English proficiency.

	Adjusted Estimate (95% CI) *
Low English proficient (ref = English proficient)	1.64 (0.67, 3.84)
Age in years	1.10 (1.01, 1.20)
Gender (ref = male)	0.80 (0.40, 1.62)

Abbreviations: OME, oral morphine equivalents; * Coefficients were also adjusted for race (White, Black, Asian, and other) but these are not reported given the small cohort size of some racial subgroups.

**Table 4 children-07-00163-t004:** Odds of no PACU pain score assessment, by English proficiency.

	Adjusted OR (95% CI) *
Low English proficient (ref = English proficient)	1.93 (0.79, 4.73)
Age in years	0.58 (0.51, 0.67)
Gender (ref = male)	0.57 (0.27, 1.19)

* Coefficients were also adjusted for race (White, Black, Asian, and other) but these are not reported given the small cohort size of some racial subgroups.

**Table 5 children-07-00163-t005:** Subgroup analysis of no PACU pain assessment by age.

Patients Age < 5 Years (N = 33)		Patients Age ≥ 5 Years (N = 224)	
	Adjusted OR (95% CI) *		Adjusted OR (95% CI) *
Low English proficient (ref = English proficient)	0.5 (0.04, 6.51)	Low English proficient (ref = English proficient)	3.23 (1.48, 7.06)
Gender (ref = male)	1.11 (0.08, 15.08)	Gender (ref = male)	0.65 (0.34, 1.24)

* Coefficients were also adjusted for race (White, Black, Asian, and other) but these are not reported given the small cohort size of some racial subgroups.

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
