# Peer review of "Language-Related Disparities in Pain Management in the Post-Anesthesia Care Unit for Children Undergoing Laparoscopic Appendectomy"

_children, 2020, doi:10.3390/children7100163_

Round 1

Reviewer 1 Report

I may have overlooked this, but I think that several factors may impact on postoperative pain (and opioid use) that were not addressed in the present study.

In a multivariate analysis assessing use of opioids it is of importance to know the amount of inflammation (CRP?, WBC?, perforation?, duration of symptoms, ALVERADO score, etc.), as well as the type of surgery (laparoscopic, open, number and size of incisions, local anaesthesia injection after procedure yes/no, etc.)

Without considering and/or correcting for these factors, I don't think the present analysis is of any value.

Specific Comments:

In the retrospective single-centre study, Dixit and collaborators assessed the effect of non-English proficiency on opioid administration and dosage in children undergoing laparoscopic appendectomy. 257 patients were included: 200 with and 57 without English proficiency.
While the study is well written and presented, some factors that might affect postoperative pain (that was not assessed directly) and opioid administration were not assessed by the authors, including severity of disease (how long were the patients sick before surgery? C-reactive protein, fever, WBC, Alvarado score, etc. before surgery? Vitals?. Likewise, its not written, if patients had perforated or non-perforated appendicitis (or maybe a negative appendectomy?) A patient with a one day history, a WBC of 10, a negative CRP and a slight inflammation at the tip of the appendix will likely have less postoperative pain, then a patient with symptoms for 5 days, a CRP above 100, fever, perforated appendicitis and significant peritonitis. Also possible injection of local anesthesia at the skin incision was not mentioned. Was a stapler or endoloops used for appendectomy? How many incisions were made? 
All these factors need to be addressed and need to be included in the analyses in the present study. In addition, the retrospective study design (prone to bias, pain score not measured, etc.) and the low number of included patients are clear disadvantages of the present study.

Reviewer 2 Report

Thank you for submitting the manuscript. I read your paper with great interest and attention. The subject of your study is interesting and of sure impact. In fact, pain therapy is a right and too often pain is undertreated, especially in the pediatric field. On the other hand, proper pain management allows to reduce complications and hospitalization.

Your paper is well written and fluent even if, in my opinion, revisions would increase the value of the paper and its readability.

  • In the results section, you write that all patients received general anesthesia and nine of them received blended anesthesia. Since the work is based on the quantity of analgesics administered in PACU, in my opinion it is necessary to better specify which type of anesthesia (inhalation, TIVA, balanced) was administered. Above all, it would be desirable to specify which and how much opioid was administered. In fact, the use of a short half-life opioid, such as remifentanil, does not provide an analgesic tail during the post-operative period, unlike other opioids.
  • In the section "opioid medications" refer to some perioperative complications and their frequency. This is important because the use of opioids, whether deficient or excessive, can cause adverse events of various kinds. In this regard, it would be better to deepen the topic of perioperative compliances (especially respiratory ones) I recommend reading and inserting as references the APRICOT study and the Italian study derived from secondary analysis. This is to underline that the use of opioids must always be associated with patient safety. In conclusion, your paper is really well written and shows sensitivity towards the theme of pain therapy, too often relegated to mere technicalities, such as the search for the perfect locoregional technique. Instead, pain therapy represents taking charge of the patient as a whole.

I hope these comments are helpful.

Kind Regards

Reviewer 3 Report

To the Authors,

The issue is interesting and the manuscript was well prepared.

The only question is that the missing of the full name of OME. Please add it in the abstract.
